# Apparent Yield Stress of Sputum as a Relevant Biomarker in Cystic Fibrosis

**DOI:** 10.3390/cells10113107

**Published:** 2021-11-10

**Authors:** Rosy Ghanem, Philippe Roquefort, Sophie Ramel, Véronique Laurent, Tanguy Haute, Tony Le Gall, Thierry Aubry, Tristan Montier

**Affiliations:** 1Univ. Brest, INSERM, EFS, UMR 1078, GGB, F-29200 Brest, France; Veronique.Laurent@univ-brest.fr (V.L.); Tanguy.Haute@univ-brest.fr (T.H.); Tony.Legall@univ-brest.fr (T.L.G.); 2Univ. Brest, IRDL UMR CNRS 6027, UFR Sciences et Techniques, 6, Avenue Victor Le Gorgeu CS 93837, CEDEX 3, 29238 Brest, France; Philippe.Roquefort@univ-brest.fr; 3Centre de Ressources et de Compétences de la Mucoviscidose, Fondation Ildys, Presqu’île de Perharidy, 29680 Roscoff, France; Sophie.Ramel@ildys.org; 4CHRU de Brest, Service de Génétique Médicale et Biologie de la Reproduction, Centre de Référence des Maladies Rares “Maladies Neuromusculaires”, F-29200 Brest, France

**Keywords:** airway mucus, rheology, biomarker, Cystic Fibrosis

## Abstract

The mucus obstructing the airways of Cystic Fibrosis (CF) patients is a yield stress fluid. Linear and non-linear rheological analyses of CF sputa can provide relevant biophysical markers, which could be used for the management of this disease. Sputa were collected from CF patients either without any induction or following an aerosol treatment with the recombinant human DNAse (rhDNAse, Pulmozyme^®^). Several sample preparations were considered and multiple measurements were performed in order to assess both the repeatability and the robustness of the rheological measurements. The linear and non-linear rheological properties of all CF sputa were characterized. While no correlation between oscillatory shear linear viscoelastic properties and clinical data was observed, the steady shear flow data showed that the apparent yield stress of sputum from CF patients previously treated with rhDNAse was approximately one decade lower than that of non-treated CF patients. Similar results were obtained with sputa from non-induced CF patients subjected ex vivo to a Pulmozyme^®^ aerosol treatment. The results demonstrate that the apparent yield stress of patient sputa is a relevant predictive/prognostic biomarker in CF patients and could help in the development of new mucolytic agents.

## 1. Introduction

Cystic Fibrosis (CF) is a genetic disorder affecting multiple organs, the lungs being those whose failure leads to premature death [1]. The pathology is due to mutations in the *cftr* gene encoding the Cystic Fibrosis Transmembrane conductance Regulator (CFTR) protein, the most frequent one being the F508del mutation. The CFTR is a channel that controls the excretion of chloride ions and negatively regulates the activity of the epithelial sodium channel (ENaC) [2]. As a consequence of mutation(s) occurring in the *cftr* gene, ionic dysregulation, causing massive water reabsorption, is responsible for the sticky dehydrated mucus covering the epithelial cells of the lung airways. This provides a favorable environment for bacteria colonization, which induces chronic inflammation and modifies the viscoelastic properties of the mucus. Moreover, CF mucus notably contains increased levels of DNA primarily derived from the disintegration of inflammatory cells such as neutrophils, leading to an increase in sputum viscosity [3]. The accumulation of a viscous mucus in the respiratory tract, combined with the bacterial infection and immune response, progressively lead to the destruction of the pulmonary parenchyma and, ultimately, to lung failure. The use of small drugs to modulate and/or potentiate the CFTR protein functions represents a promising strategy for the treatment of CF patients. The most recent therapy approved by the FDA and EMA is a tritherapy combining ivacaftor, elexacaftor and tezacaftor (Kaftrio^®^, Vertex Pharmaceuticals, Boston, MA, USA) [4,5]. However, this approach is mutation-dependent and therefore cannot be applied to all CF patients. Additionally, some patients have developed adverse effects, leading to the interruption of their pharmacological treatment [6,7]. Gene therapy is also a promising alternative to CF protein treatments as it is independent of mutation profiles and could thus offer real benefit to all patients; however, long-term tolerance is still unknown. To administrate such treatments, aerosol delivery represents an ideal method to target the pulmonary epithelium as it is a non-invasive, loco-regional administration method that allows us to bypass the hepatic first-pass effect. The latest non-viral gene therapy trial was performed using aerosol delivery of a formulation based on the cationic lipid GL67 [8]. Although an increase of 3.7% in the FEV1 (forced expiratory volume in one second) was observed during the clinical study, no significant clinical improvement was recorded in patients. This disappointing result may be partly explained by the presence of the CF mucus covering the respiratory tract, which impedes nanolipoplexes (as well as recombinant viruses) from reaching the underlying target epithelium [9]. In this respect, mucus is a key barrier that gene delivery has to overcome in order to be successful. The role played by mucus as a complex fluid, i.e., a structured medium with complex rheological properties, has attracted much attention for a long time [10,11,12]. For instance, a rheological study performed on sputa from CF patients treated with rhDNAse (Pulmozyme^®^), which is currently the main mucolytic treatment indicated before a physiotherapy session, has shown a reduction in the sputum viscosity, bringing a clinical benefit [13]. Bacterial infection was also proven to influence the rheological behavior of the mucus by increasing both its storage modulus and Newtonian viscosity [14]. A very recent study suggests that a critical stress, defined as the stress at which the storage (elastic) modulus is equal to the viscous (loss) modulus, could be a possible marker of chronic diseases [15]. However, in their paper, the authors used a homogenization method prior to rheological characterization, which could have impacted the rheological properties of the patients’ sputa. Moreover, the critical stress that the authors defined was obtained in a non-linear regime, where a Fourier analysis is needed [16], meaning that the storage and loss moduli used to define the critical stress are only part of the response. In the present work, we aim at studying the linear and non-linear rheological properties of non-pre-sheared CF sputa, using both linear oscillatory shear and steady shear flow, in order to determine a biophysical marker of CF patient mucus rheology.

## 2. Materials and Methods

### 2.1. Sputum Collection

Sputa from CF patients were collected at the “Centre de Ressources et de Compétences de la Mucoviscidose (CRCM)” in Roscoff (France). Patients, or the parents of underage patients, were informed about the purpose of the study. Sputum collection was considered clinical waste and no patient intervention was necessary for this study. For a given patient, sputum was obtained either by spontaneous expectoration or following Pulmozyme^®^ induction. Patient clinical data were anonymized and kept confidential; they were correlated to rheological measurements. Sputa were stored at −20 °C and then carried on ice to the laboratory, where they were stored at −20 °C prior to rheological testing. Non-CF mucus was obtained by washing (with 5 mL of 0.9% NaCl) an endotracheal tube used during the heart surgery of a patient without lung disease. This material as well as its content were considered clinical waste and no additional procedure was necessary for the purpose of this study. After washing, the mucus was stored at −20 °C before rheological experiments.

### 2.2. Patient Demographics

This study included sputa from 20 subjects (10 men and 10 women) with an average age of 32 ± 8 years at the time of sputum collection (Table 1). The oldest subject was 58 years old while the youngest was 5 years old. All patient sputa were found colonized by bacteria, 18 containing *Pseudomonas aeruginosa*. Only four patients experienced pulmonary exacerbations during the study period. The average FEV1 was 50% ± 16%, corresponding to moderate to severe pulmonary insufficiencies.

### 2.3. Rheological Experiments

Plane simple shear rheological experiments were carried out on ~0.6 mL sputa samples at 37 °C with a MCR702 Anton Paar rheometer. Two types of rheometrical tests were performed: oscillatory shear tests were used to characterize the linear viscoelastic moduli G′ (storage modulus) and G″ (loss modulus), and steady shear tests were used to characterize the flow behavior (apparent viscosity η as a function of shear stress τ) of sputa. The viscoelastic and flow properties of all samples were measured using a cone and plate geometry (diameter: 50 mm, cone angle: 1°, truncation: 100 µm); sandblasted parallel plates (diameter 25 mm, gap: 1 mm) were also used to investigate possible slip effects. For all sputa tested, a thin layer of low-viscosity silicone oil was placed at the air/sample interface in order to prevent evaporation. The characterization of the linear viscoelastic properties was performed using a two-step protocol: first, a strain sweep experiment at a fixed angular frequency of 1 rad.s^−1^, allowing us to determine the linear viscoelastic regime (sinusoidal response and stress proportional to the magnitude of the applied strain), followed by a frequency sweep from 100 to 0.1 rad.s^−1^, at a fixed strain amplitude of 1%, which lay within the linear regime for all samples tested. Flow curves of all samples were obtained using shear rate sweep experiments, from 0.01 to 1000 s^−1^. For the more viscous samples, creep experiments in the linear response regime were also performed in order to obtain the zero-shear Newtonian viscosity.

### 2.4. Ex Vivo Treatment of Mucus

A sample of sputum previously collected from a non-induced patient (p18) was thawed at room temperature and placed on a Petri dish inside an exposure box (12 × 8.3 × 6.7 cm). Pulmozyme^®^ (rhDNAse, 2500 IU/2.5 mL, Roche) was aerosolized using the eFlow^®^ Rapid Nebulizer (PARI, France). Rheological characterization was performed immediately after this treatment. Furthermore, Istendo^®^ (N-acetyl-cysteine, 1 g/5 mL, Laboratoires Delbert) was deposited on another sample of p18 sputum and directly analyzed in rheology.

### 2.5. Statistical Analyses

Data were analyzed using GraphPad Prism. To assess statistical difference, the non-parametric Mann–Whitney test was performed. Differences were considered statistically significant for *p* values < 5%.

## 3. Results

### 3.1. Preliminary Rheometrical Results

In the present work, sputa were harvested from 20 CF patients either by spontaneous expectoration or after rhDNAse induction. For each patient, the *cftr* mutation as well as the FEV1 and the bacterial infections were recorded (Appendix A). All sputa were stored at −20 °C prior to rheological characterization [8]. As previous studies have suggested various preparation standardized protocols, such as saliva removal or vortex homogenization, in order to homogenize samples prior to testing [15,17], the possible effect of sample centrifugation on the rheological behavior was first investigated by centrifugation at 2000× *g* for 10 min. In the present work, no significant effect of centrifugation on the rheological response was ever observed (Figure 1); thus, all sputa were used per se, without any preparation prior to rheological measurements, which allowed us to eliminate any artefacts. Moreover, repeatability was shown to be satisfactory, as assessed with samples for which a sufficient volume was available (Figure 1). Lastly, using parallel plates with different roughness, we demonstrated that no significant slip occurred during the flow experiments (Appendix A). However, it is worth pointing out that the volume of some patients’ sputa was too small to allow complete rheological characterization, including repeatability, slip effects and centrifugation effects.

### 3.2. Linear and Non-Linear Rheological Characterization

For all samples, the viscoelastic moduli G′ and G″ were first studied as a function of strain in order to determine the extent of the linear viscoelastic regime (Appendix A). Then, frequency sweep experiments were carried out at a 1% strain amplitude, chosen in the linear viscoelastic domain. G′ and G″, as well as G″/G′ at 0.1 rad.s^−1^ and at 10 rad.s^−1^, for all samples are given in Table 2. No correlation between clinical data and G′, G″ data was observed. However, as highlighted in Figure 2A, standard deviations of both G′ and G″ were noticeably much lower in the rhDNAse induction group. Using exactly the same experimental procedure for all samples, i.e., shear rate sweep experiments, the flow curves of all sputa were also characterized. They all exhibited the same features: a Newtonian behavior at low stresses, followed by a very sharp decrease in the apparent viscosity, which appeared to be a near discontinuity. The stress threshold in the non-linear part of the flow curve can be identified with an apparent yield stress (τ_γ_). It is worth pointing out that τ_γ_ is not a parameter characterizing a material at a given temperature and pressure but, rather, a protocol-dependent threshold parameter characterizing the stress needed to make a material flow [18]. The Newtonian viscosity and apparent yield stress for all samples are given in Table 2. The presence of an apparent yield stress in the flow curve of sputa was ascribed to the shear-induced fracture of a transient network, relatively similar to what is observed in associative polymers [19,20,21]. Most remarkably, two groups of patients can be distinguished when considering the value of the apparent yield stress: one with a yield stress of approximately 1 Pa, which is also observed for non-CF mucus, and the other one with an apparent yield stress approximately one decade higher (Figure 2B). Most interestingly, sputa inducted with rhDNAse, as well as non-CF sputum, belong to the lower yield stress group, whereas spontaneous sputa correspond to the higher yield stress group, except for three patients (Table 2 and Figure 2). Moreover, comparison of the standard deviations shows that the yield stress dispersion is much lower in the rhDNAse induction group than in the spontaneous sputum group. In addition, data in Figure 2B show that the Newtonian viscosity level of both groups is different, the treated group having, on average, a slightly lower Newtonian viscosity than the non-treated one, as previously observed [13,22]. However, the difference between the two groups is significantly less pronounced when considering the Newtonian viscosity, and the variability in viscosity is also much greater than that in yield stress for non-treated patients. In this respect, the apparent yield stress appears to be a much more sensitive and accurate rheological parameter than viscosity to characterize the effect of rhDNAse induction.

### 3.3. Ex Vivo Sputum Treatment

In order to confirm the role of rhDNAse, the sputum previously harvested from a non-conditioned patient was treated with an aerosol of rhDNAse. The results obtained under in vitro conditions (Figure 3) clearly confirm the effect of rhDNAse on the apparent yield stress. Moreover, we wondered whether N-acetyl-cysteine (NAC), which only affects the mucin network [23], could also induce a decrease in the apparent yield stress. As shown in Figure 3, ex vivo NAC treatment of the same sputum leads to a much smaller decrease in the apparent yield stress than rhDNAse treatment.

## 4. Discussion

The major role played by extracellular DNA in the structure of CF mucus could explain the effect of rhDNAse, which only and directly affects the DNA/actin network, on the apparent yield stress. Indeed, the amount of extracellular DNA in CF mucus is much higher than in non-CF mucus [24,25]. The mucus structure is quite complex, consisting of a network involving interactions between mucin and non-mucin proteins, forming the mucus scaffold [26]. Extracellular DNA contributes to the structural and viscoelastic properties of mucus by forming bundles with F-actin, which are connected within a network, consolidating the mucin network [27,28,29,30]. In addition, due to chronic infections, the presence of bacterial biofilm, non-homogeneously distributed within the sputum, has to be a priori considered, as it is known to also exhibit viscoelastic properties [31,32,33]. Treatment with rhDNAse affects the DNA/actin network: it leads to a reduction in both the concentration and size of extracellular DNA, which is responsible for the fracture of the DNA/actin network [34,35]. The small effect of NAC, which only affects the mucin network, on the apparent yield stress indirectly confirms that the higher apparent yield stress is due to the breakdown of the DNA/actin network. This interpretation is also supported by the fact that rhDNAse has virtually no effect on CF biofilm [36] or on the mucin network [37]. Moreover, as non-CF sputum is devoid of biofilm, but still exhibits an apparent yield stress of the same order of magnitude as that of treated sputum, we propose that the low apparent yield stress can be attributed to the destruction of the mucin network. The overall results of the present study therefore strongly suggest that the higher apparent yield stress is a rheological signature of the destruction of the DNA/actin network, whereas the lower apparent yield stress is a rheological signature of the mucin network. In order to give a summary picture, a schematic illustration of the structure of CF sputum, as well as the effect of rhDNAse and NAC, is given in Figure 4.

## 5. Conclusions

In the present study, linear viscoelastic properties (storage modulus G′ and loss modulus G″), as well as flow properties (Newtonian viscosity, yield stress), of CF sputa were characterized. Interestingly, the apparent yield stress, rather than the linear viscoelastic moduli G′ and G″ and even the Newtonian viscosity, turned out to be the most relevant biomarker for the development and the monitoring of mucolytic agents acting on the DNA/actin network. This could also be used as a key parameter to study the efficiency of new pharmacological therapies such as Trikafta^®^ or prior to gene therapy delivery, as well as in the development of in vitro mucus models for the screening of new drugs or the improvement of their formulations [38,39].

## Figures and Tables

**Figure 1 cells-10-03107-f001:**
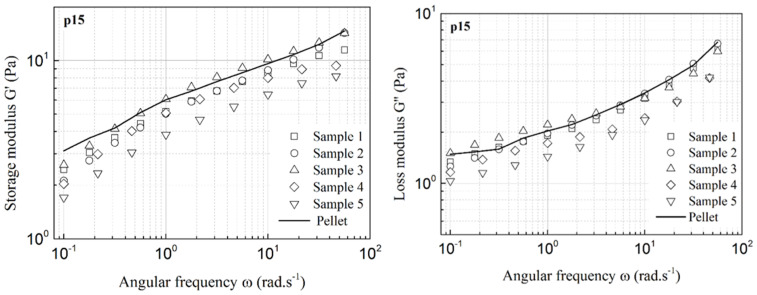
Repeatability and effect of centrifugation: storage modulus (G′) and loss modulus (G″) for 6 samples taken from the same sputum (obtained from patient p15); sample 6 is the pellet obtained after centrifugation at 2000× *g*, for 10 min.

**Figure 2 cells-10-03107-f002:**
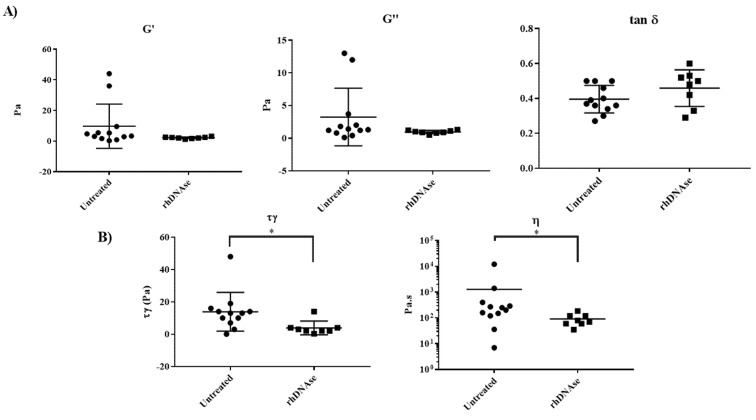
(**A**) Graphical comparison of rheological data obtained in oscillatory and steady shear flow of mucus from spontaneous or induced CF expectoration collected for this study. Storage modulus (G′) and loss modulus (G″) data were measured at a strain amplitude of 1% and at an angular frequency of 0.1 rad.s^−1^ and tanδ = G″/G′. (**B**) Graphical comparison of apparent yield stress (τ_γ_) and Newtonian viscosity (η) of untreated and rhDNAse-treated mucus. Statistical analysis with the non-parametric Mann–Whitney test was performed to compare τ_γ_ and η values in patients from the two groups; the symbol “*” denotes a *p* value < 5%.

**Figure 3 cells-10-03107-f003:**
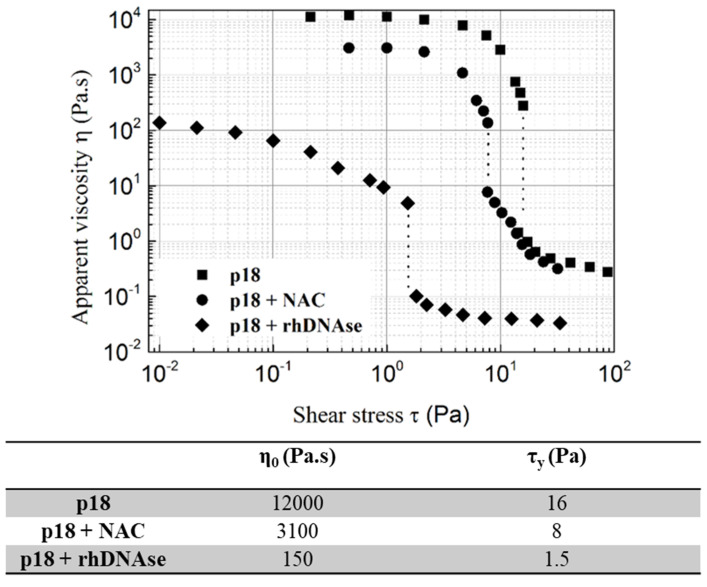
Flow curves, Newtonian viscosity (η_0_) and apparent yield stress (τ_γ_) of sputum collected from p18 (collected without rhDNAse induction) and treated or not with either rhDNAse or N-acetyl-cysteine (NAC). The apparent yield stress is defined as the stress at the flow curve near discontinuity.

**Figure 4 cells-10-03107-f004:**
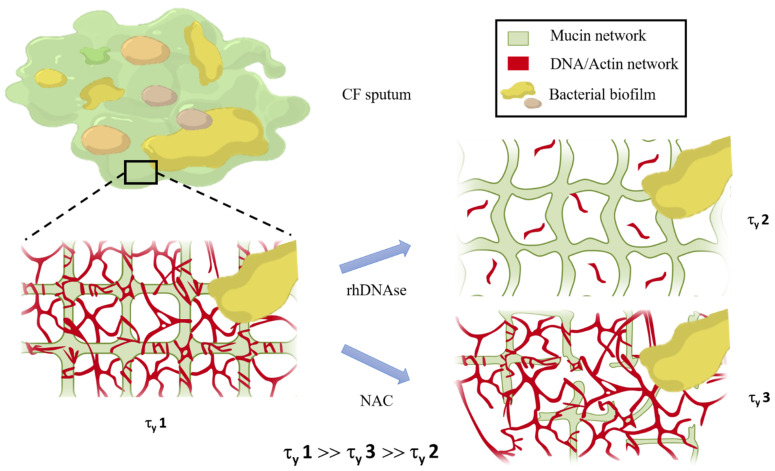
Schematic representation of CF sputum composed of two distinct networks and bacterial biofilm. Mucin and DNA/actin networks are interconnected. Untreated CF sputum exhibits high yield stress (τ_γ_ 1). When CF sputum is treated with rhDNAse, the mucin network is preserved but DNA/actin network is broken, leading to the decrease in the apparent yield stress to the same level as that of non-CF sputum (τ_γ_ 2). On the other hand, when CF sputum is exposed to NAC (N-acetyl-cysteine), some of the disulfide bridges of the mucin network are broken, but the DNA/actin network is largely preserved, resulting in a slightly lower decrease in the yield stress (τ_γ_ 3).

**Table 1 cells-10-03107-t001:** Clinical data of CF patients at the time of sputum collection.

Patient	Birth Year	Gender *	Mutation	FEV1	Bacteria	Sputum Induction
**1**	1988	W	F508 del/F508 del	55%	*Achromobacter* spp.	No
**2**	2000	M	F508 del/F508 del	88%	MSSA ** *Staphylococcus aureus*	No
**3**	1993	M	357 del C/357 del C	30%	*P. aeruginosa*	No
**4**	1961	M	N1303K/1898 + 5 G > A	46%	*P. aeruginosa* *Escherichia coli*	rhDNAse
**5**	1978	M	W 1282 X/R117H	73%	MSSA *P. aeruginosa*	rhDNAse
**6**	1985	W	F508 del/G542X	30%	MSSA *P. aeruginosa* *Stenotrophomonas maltophilia*	No
**7**	1991	W	F508 del/F508 del	49%	MSSA *P. aeruginosa*	No
**8**	1996	W	F508 del/F508 del	35%	*Achromobacter* spp. *P. aeruginosa*	rhDNAse
**9**	1987	W	F508 del/F508 del	30%	MRSA *** *P. aeruginosa*	No
**10**	1975	W	F508 del/F508 del	53%	*Burkholderia cenocepacia*	No
**11**	1987	W	F508 del/91 (G-R)	66%	*P. aeruginosa*	rhDNAse
**12**	1974	M	F508 del/F508 del	45%	*P. aeruginosa*	rhDNAse
**13**	2006	M	F508 del/F508 del	93%	*Haemophilus influenza SSA*	rhDNAse
**14**	1989	W	F508 del/w882x	31%	*P. aeruginosa* *Candida albicans*	No
**15**	1979	M	F508 del/F508 del	30%	MSSA *P. aeruginosa*	rhDNAse
**16**	1980	M	F508 del/G411X	53%	MSSA *P. aeruginosa*	No
**17**	1985	W	F508 del/F508 del	23%	*Aspergilus fumigatus* *P. aeruginosa*	rhDNAse
**18**	1982	M	F508 del/F508 del	35%	*P. aeruginosa*	No
**19**	1985	W	F508 del/F508 del	70%	MSSA *P. aeruginosa*	No
**20**	2014	M	F508 del/4005 G > A	114%	*H. influenza* MSSA	No

* M = man and W = woman, ** MSSA = Methicillin-Susceptible *Staphylococcus aureus*, *** MRSA = Methicillin-Resistant *Staphylococcus aureus*.

**Table 2 cells-10-03107-t002:** Rheological characterization of CF sputa collected from patients who were previously treated, or not, with rhDNAse. The sputum obtained from a non-CF patient (collected by washing an endotracheal tube used during heart surgery) was used a reference. G′ and G″ data were measured at a strain amplitude of 1% and at an angular frequency of 0.1 rad.s^−1^ and 10 rad.s^−1^.

Patient	Pretreatment	Oscillatory Shear						Steady Shear	
		0.1 rad.s^−1^			10 rad.s^−1^				
		G′ (Pa)	G″ (Pa)	tanδ (G″/G′)	G′ (Pa)	G″ (Pa)	tanδ (G″/G′)	η (Pa.s)	τ_y_ (Pa)
**1**	No	5.3	1.8	0.34	10.3	2.5	0.24	290	13
**2**	No	36	13	0.36	81	19	0.23	1400	48
**3**	No	2.8	1.3	0.46	6.7	1.8	0.27	150	10
**4**	rhDNAse	1.2	0.5	0.42	2.6	0.75	0.29	35	2
**5**	rhDNAse	3.1	0.9	0.29	6.5	1.7	0.26	185	14
**6**	No	3.3	1.2	0.36	6.9	1.8	0.26	160	19
**7**	No	5.4	2	0.37	12.2	3.1	0.25	270	14
**8**	rhDNAse	2.0	1	0.50	4.6	1.4	0.30	70	4
**9**	No	0.2	0.1	0.50	0.7	0.36	0.51	7	0.1
**10**	No	9.5	3.7	0.39	20	4.5	0.23	400	14
**11**	rhDNAse	2.3	1.1	0.48	6.2	2	0.32	60	2
**12**	rhDNAse	1.6	0.9	0.56	4.5	1.5	0.33	60	3
**13**	rhDNAse	2.4	0.8	0.33	5.2	1.3	0.25	120	4
**14**	No	3.0	1.2	0.40	6.7	1.7	0.25	120	10
**15**	rhDNAse	2.5	1.3	0.52	8.6	3.1	0.36	80	2
**16**	No	4.7	1.4	0.30	10.7	2.3	0.21	200	13
**17**	rhDNAse	2.0	1.2	0.60	9.7	3.4	0.35	120	0.3
**18**	No	44	12	0.27	93	18	0.19	12,000	16
**19**	No	1.6	0.8	0.50	3.7	0.9	0.24	250	7
**20**	No	0.8	0.4	0.50	2.3	0.8	0.35	36	3
**non-CF**	No	0.5	0.3	0.60	1.8	0.8	0.44	20	0.3

## Data Availability

Not applicable.

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
