# Peer review of "Apparent Yield Stress of Sputum as a Relevant Biomarker in Cystic Fibrosis"

_cells, 2021, doi:10.3390/cells10113107_

Round 1

Reviewer 1 Report

The manuscript presents the study of the linear and non-linear rheological properties of non-pre-sheared CF sputa. The authors applied both linear oscillatory shear and steady shear flow, in order to characterize the CF sputa. The rheological parameters of the sputum can be good indicators of the condition of the CF patient and for the evaluation or prediction of the effectiveness of the treatments. The article is well-structured, adequate method is used in order to characterize CF sputum.

Reviewer’s comments concerning the manuscript are the follows:

  1. Please detail the rheological method in the method part of the article, specify the exact geometries of the measuring systems and gap height used. Please give the details of the strain, and frequency sweeps, and the flow curves (shear rate range, steps, frequency range, strain range, LVE range, proposal strain for frequency sweeps etc.).
  2. Please change Table S1 to Table 1 (line 130).
  3. The lines 130-134 contain repetitions.
  4. Please specify the centrifugation method! Has the supernatant been removed? Sputum samples show large differences depending on the patient and the patient’s condition. The authors presented the centrifugation had no effect on the rheological characteristics of sputum, but this statement can be true for all samples examined?
  5. The authors also used different measuring devices (CP50, PP/S 25) for the methodology setting, and found that the geometry of the measuring system had no effect on the results. Which measuring device was used for further measurements?
  6. How was it ensured that the sample did not dry out/evaporate during the rheological measurements?
  7. The applied 1% strain value seems too large, how it was applicable to the less consistent sputum samples? Figure S2 presented the amplitude sweep of the sputum of the patient No15. Can this sputum sample be considered as the less consistent sample respect to the LVE range? In the description of Table 2 (line 183), the strain value is 0.1%, while in the text (line 150) it is 1%, please revise it.
  8. Why did the author present the G’ and G” at 0.1 rad/s? Sputum samples can show high frequency dependency, it would be advised to present the moduli at other angular frequency as well (e.g. 1 or 10 rad/s)

Author Response

Dear Reviewer,

Below please find our point-by-point response to your comments, including the modifications made to the manuscript.

  1. Please detail the rheological method in the method part of the article, specify the exact geometries of the measuring systems and gap height used. Please give the details of the strain, and frequency sweeps, and the flow curves (shear rate range, steps, frequency range, strain range, LVE range, proposal strain for frequency sweeps etc.).

All details regarding the rheological methods used in the present study were given in the method part of the revised manuscript:

2.3. Rheological experiments

Plane simple shear rheological experiments were carried out on ~0.6mL sputa samples at 37°C with a MCR702 Anton Paar rheometer. Two types of rheometrical tests were performed: oscillatory shear tests were used to characterize the linear viscoelastic moduli G’ (storage modulus) and G” (loss modulus), and steady shear tests were used to characterize the flow behavior (apparent viscosity η as a function of shear stress τ) of sputa. The viscoelastic and flow properties of all samples were measured using a cone and plate geometry (diameter: 50mm, cone angle: 1°, truncation: 100µm); sandblasted parallel plates (diameter 25mm, gap: 1mm) were also used to investigate possible slip effects. For all sputa tested, a thin layer of low-viscosity silicone oil was put on the air/sample interface in order to prevent evaporation. The characterization of the linear viscoelastic properties was performed using a two-step protocol: first, a strain sweep experiments at a fixed angular frequency of 1 rad.s-1, allowing to determine the linear viscoelastic regime (sinusoidal response and stress proportional to the magnitude of the applied strain) then a frequency sweep from 100 to 0.1 rad.s-1, at a fixed strain amplitude of 1% which lies within the linear regime for all samples tested. Flow curves of all samples were obtained using shear rate sweep experiments, from 0.01 to 1000s-1. For the more viscous samples, creep experiments in the linear response regime were also performed in order to get the zero-shear Newtonian viscosity.

  1. Please change Table S1 to Table 1 (line 130).

This was changed in the revised version of the manuscript.

  1. The lines 130-134 contain repetitions.

We agree: the sentence "Both oscillatory and steady shear measurements were carried out in order to study the linear viscoelastic (storage modulus G’ and loss modulus G” as a function of strain and frequency) and flow properties (viscosity as a function of stress) of sputa" was removed from the revised version of the article.

  1. Please specify the centrifugation method! Has the supernatant been removed? Sputum samples show large differences depending on the patient and the patient’s condition. The authors presented the centrifugation had no effect on the rheological characteristics of sputum, but this statement can be true for all samples examined?

As precised in the legend of Figure 1, some samples were centrifuged at 2000 g for 10 min in order to study the effect of sample preparation. Of course, the supernatant was removed prior to rheological characterization of the centrifugation pellet. Unfortunately, due to the small amount of sputum for some patients, the effect of centrifugation could not be studied systematically. This was precised in the revised version of the manuscript.

  1. The authors also used different measuring devices (CP50, PP/S 25) for the methodology setting, and found that the geometry of the measuring system had no effect on the results. Which measuring device was used for further measurements?

This was precised in part 2.3 of the revised manuscript.

  1. How was it ensured that the sample did not dry out/evaporate during the rheological measurements?

As mentioned in part 2.3 of the revised article, a thin layer of low-viscosity silicone oil was put on the air/sample interface in order to prevent evaporation.

  1. The applied 1% strain value seems too large, how it was applicable to the less consistent sputum samples? Figure S2 presented the amplitude sweep of the sputum of the patient No15. Can this sputum sample be considered as the less consistent sample respect to the LVE range? In the description of Table 2 (line 183), the strain value is 0.1%, while in the text (line 150) it is 1%, please revise it.

The amplitude of the oscillatory strain is 1% for all samples. The error in the description of Table 2 was corrected. For all sputa, we carefully verified that 1% was within the linear viscoelastic regime; for the less consistent samples it is indeed very close to the critical strain.

  1. Why did the author present the G’ and G” at 0.1 rad/s? Sputum samples can show high frequency dependency, it would be advised to present the moduli at other angular frequency as well (e.g. 1 or 10 rad/s)

G’, G” as well as G”/G’ are presented at 0.1 rad/s and 10 rad/s in Table 2 of the revised manuscript.

Reviewer 2 Report

This article provides a study of the rheology of sputa collected from CF patients, by performing several rheological measurements and shows that the apparent yield stress of patient sputa can be considered as a relevant marker for CF patients. 

The scientific topic is of high relevance, the methology is coherent and the artcile is well written and easy to follow. I recommend for publication Cells, if the following minor suggestions are successfully taken into account:

-section 2.3: It would be better to indicate the sample volumes, although they can be computed using the diameters of both parallel plate and cone plate geometries and the measurement gaps.  Indeed, the sample volume is critical for measurements on real mucus, as very small volumes of mucus can be associated with high errors.

-Figure 1 characterizes the repeatability of the rheological measurements and aims at demonstrating that the centrifugation has no effect on the rehological measurements. More quantitative agruments could be given to confirm this point. Additionaly, error bars could be added to the figure, which would give more credibility on the conlusions. 

- Could you give more details on the choice of using tau_flow instead of tau_y ?

- Why do you use different samples for the measurements of figures S1 and S2 ? This would help comparing the values of tau_y and tau_flow.

- Why is there a difference beteen the two curves ? (as it does not come from a slip effect). Is it directly linked to the reproductibility error when using two samples from the same source and two different geometries? The difference seems to be rather large to me

Author Response

Dear Reviewer,

Below please find our point-by-point response to your comments, including the modifications made to the manuscript.

-section 2.3: It would be better to indicate the sample volumes, although they can be computed using the diameters of both parallel plate and cone plate geometries and the measurement gaps.  Indeed, the sample volume is critical for measurements on real mucus, as very small volumes of mucus can be associated with high errors.

These precisions were given in part 2.3 of the revised manuscript.

-Figure 1 characterizes the repeatability of the rheological measurements and aims at demonstrating that the centrifugation has no effect on the rheological measurements. More quantitative arguments could be given to confirm this point. Additionally, error bars could be added to the figure, which would give more credibility on the conclusions. 

As precised in the revised manuscript, it was impossible to study the effect of centrifugation and the repeatability on all samples because the volume of some patients sputa was too small to allow a complete rheological characterization, but we think that the results plotted in Figure 1 are still meaningful. Moreover, the rheological data plotted in Figure 1 give an order of magnitude of the experimental error.

- Could you give more details on the choice of using tau_flow instead of tau_y ?

First, the stress ty discussed in the manuscript is an apparent yield stress, and not a true yield stress, since all sputa tested in the present work exhibit a Newtonian zero-shear viscosity, meaning that they flow even at very low shear rates below ty.  The apparent yield stress discussed in the manuscript is the stress corresponding to a drastic decrease of the apparent viscosity. As discussed in the introduction, ty is a priori quite different from the stress at which G’=G” ; this stress which could appear as a flow stress is but very difficult to discuss as it is defined in the non-linear-viscoelastic regime where higher harmonics are needed, so that it is very difficult to see what it really means.

- Why do you use different samples for the measurements of figures S1 and S2? This would help comparing the values of tau_y and tau_flow.

As discussed in the response to your previous question, the point is not to compare tau y and tau flow. Figure S2 is only used in order to determine the extent of the linear viscoelastic regime; the non-linear viscoelastic properties, which would need a Fourier analysis, have not been considered in this work.

- Why is there a difference between the two curves? (as it does not come from a slip effect). Is it directly linked to the reproducibility error when using two samples from the same source and two different geometries? The difference seems to be rather large to me

Indeed, the difference is mainly due to the use of two different samples from the same source, but also to the use of two different geometries: a cone and plate for which shear is homogeneous, and a parallel plate, for which it is not. Anyway, the point is that for both geometries, a drastic drop of the apparent viscosity at comparable shear stresses is observed, meaning that such an abrupt decrease cannot be attributed to slip effects, but is the rheological signature of an apparent yield stress.

Round 2

Reviewer 1 Report

All corrections have been made, the manuscript is acceptable in its current form.